

# Putative carboxylesterase gene identification and their expression patterns in *Hyphantria cunea* (Drury)

Jia Ye[1,*], Dingze Mang[2,*], Ke Kang[1,3], Cheng Chen[1], Xiaoqing Zhang[1], Yanping Tang[1], Endang R. Purba[4], Liwen Song[5], Qing-He Zhang[6] and Longwa Zhang[1]

[1] Anhui Provincial Key Laboratory of Microbial Control, Engineering Research Center of Fungal Biotechnology, Ministry of Education, School of Forestry & Landscape Architecture, Anhui Agricultural University, Hefei, China

[2] Graduate School of Bio-Applications and Systems Engineering, Tokyo University of Agriculture and Technology, Tyoko, Japan

[3] Anhui Forestry Bureau, Hefei, China

[4] Structural Cellular Biology Unit, Okinawa Institute of Science and Technology Graduate University, Okinawa, Japan

[5] Jilin Provincial Academy of Forestry Sciences, Changchun, China

[6] Sterling International, Inc., Spokane, WA, USA

[*] These authors contributed equally to this work.

Corresponding author
Longwa Zhang,
zhanglw@ahau.edu.cn,
longwazhang@126.com

## ABSTRACT

The olfactory system of insects is important for behavioral activities as it recognizes internal and external volatile stimuli in the environment. Insect odorant degrading enzymes (ODEs), including antennal-specific carboxylesterases (CXEs), are known to degrade redundant odorant molecules or to hydrolyze important olfactory sex pheromone components and plant volatiles. Compared to many well-studied Type-I sex pheromone-producing lepidopteran species, the molecular mechanisms of the olfactory system of Type-II sex pheromone-producing *Hyphantria cunea* (Drury) remain poorly understood. In the current study, we first identified a total of ten CXE genes based on our previous *H. unea* antennal transcriptomic data. We constructed a phylogenetic tree to evaluate the relationship of HcunCXEs with other insects' CXEs, and used quantitative PCR to investigate the gene expression of *H. cunea* CXEs (HcunCXEs). Our results indicate that HcunCXEs are highly expressed in antennae, legs and wings, suggesting a potential function in degrading sex pheromone components, host plant volatiles, and other xenobiotics. This study not only provides a theoretical basis for subsequent olfactory mechanism studies on *H. cunea,* but also offers some new insights into functions and evolutionary characteristics of CXEs in lepidopteran insects. From a practical point of view, these HcunCXEs might represent meaningful targets for developing behavioral interference control strategies against *H. cunea*.

## INTRODUCTION

A complete insect olfactory process requires the participation and cooperation of various olfaction-related proteins (*Scott et al., 2001*; *Vogt, 2003*; *Leal, 2013*). During the process, external liposoluble odor molecules first pass through the polar pores on the sensillum surface, then enter the lymph under the integument where they further combine with odorant binding proteins (OBPs) before being transferred to the dendritic membrane of olfactory receptor neurons (ORNs) (*Tegoni, Campanacci & Cambillau, 2004*; *Leal, 2013*; *Pelosi et al., 2018*). The molecule-bound odorant receptors (ORs) then convert the chemical signals into electrical signals that are transmitted to the central nervous system through axons of the ORNs (*Song et al., 2008*). This whole process guides insects to make relevant physiological responses and behavioral decisions. Once the signal transmission is completed, redundant odorant molecules need to be degraded or inactivated by odorant degrading enzymes (ODEs) in the antennal sensilla; otherwise, the odorant receptors will remain in a stimulated state, which may lead to poor spatio-temporal resolution of the odor signal, and pose fatal hazards to the insects (*Vogt & Riddiford, 1981*; *Steinbrecht, 1998*; *Durand et al., 2010b*; *Leal, 2013*). ODEs degrade redundant odorant molecules in the lymph of antennal sensilla and within the cells (*He et al., 2014a*). Traditionally, ODEs can be divided into five categories based on the structural difference of various target substances: carboxylesterase (CXE), cytochrome P450 (CYP), alcohol dehydrogenase (AD), aldehyde oxidase (AOX) and glutathione *S*-transferase (GST) (*Rybczynski et al., 1989*; *Ishida & Leal, 2005*; *Pelletier et al., 2007*; *Durand et al., 2010a*). However, ODEs of different categories have been shown to catalytically interact with odor molecules of the same type and structure. It is currently believed that the different enzyme families of ODEs may work together in degradation and clearing of the same type of odor molecule (*Steiner, Chertemps & Mabéche, 2019*).

As primary metabolic enzymes, CXEs are widely distributed among insects, microbes and plants (*Guo & Wong, 2020*). The active site contains several conserved serines, which promote the cleavage and formation of ester bonds (*Bornscheuer, 2002*) and play an important role in the metabolism of heterologous substances, pheromone degradation, neurogenesis, developmental regulation and many other functions (*Yu et al., 2009*). In addition to the metabolism and detoxification of endobiotics and xenobiotics, another important role of CXEs is to maintain the sensitivity of ORNs. The CXEs enable rapid degradation of stray odors and prevent vulnerable ORNs from being continuously invaded by harmful volatile xenobiotics (*Li et al., 2013*). So far, a large number of genes encoding CXEs have been identified and their functions in insect olfaction have also been investigated in various insects, including *Drosophila melanogaster*, *Mamestra brassicae*, *Antheraea polyphemus*, *Sesamia nonagrioides*, *Popillia japonica*, *Spodoptera littoralis*, *Epiphyas postvittana*, *Agrilus planipennis*, *S. litura*, *S. exigua*. (*Vogt, Riddiford & Prestwich, 1985*; *Maibeche-Coisne et al , 2004*; *Ishida & Leal, 2005*; *Merlin et al., 2007*; *Ishida & Leal, 2008*; *Jordan et al., 2008*; *Durand et al., 2010b*; *Mamidala et al., 2013*; *He et al., 2014a*; *He et al., 2014b*; *He et al., 2014c*; *He et al., 2015*; *Chertemps et al., 2015*). For instance, the *A. polyphemus* pheromone-degrading enzyme CXE (*Apol* PDE) was shown

to effectively degrade its sex pheromone acetate component (*Maibeche-Coisne et al , 2004*; *Ishida & Leal, 2005*). In *P. japonica* and *D. melanogaster*, the purified native or recombinant antennal CXEs were found to degrade their sex pheromone constituents (*Ishida & Leal, 2008*; *Younus et al., 2014*). In addition, some of CXEs from *S. exigua*, *S. littoralis* and *S. litura* were also found to degrade both their sex pheromones and plant volatiles, as well as hydrolyze volatile esters released from their natural food sources (*Gomi, Inudo & Yamada, 2003*; *Durand et al., 2011*; *Chertemps et al., 2015*).

The fall webworm, *Hyphantria cunea* (Drury) (Lepidoptera; Erebidae), native to North America, is a worldwide quarantine pest insect. This moth has now spread to most European countries (except the Nordics), South Korea, North Korea and China, and lately to Central Asia (*Itô & Miyashita, 1968*; *Gomi, 2007*). As an invasive pest, *H. cunea* was first found in Dandong (Liaoning province, China) and has rapidly spread to Hebei and adjacent provinces in China (*Gomi, 2007*; *Yang et al., 2008*; *Tang, Su & Zhang, 2012*). In 2012, the State Forestry Administration's Forest Pest Inspection and Identification Center identified the first outbreak of *H. cunea* in Sanshan district, Wuhu City, Anhui Province, which was the southernmost known outbreak of *H. cunea*. Its invasion has caused serious damage to local forests, agricultural crops and landscaping/ornamental trees, resulting in great economic and ecological losses. Thus, effective quarantine programs and environmentally safe pest management solutions are needed to combat this serious invasive pest insect. More importantly, a better understanding of its chemical ecology may facilitate more effective pest management strategies. Previous studies have described four sex pheromone components, including two straight chain aldehydes, (9Z,12Z)-octadecadienal (Z9, Z12-18Ald) and (9Z,12Z,15Z)-octadecatrienal (Z9, Z12, Z15-18Ald), and two epoxides, (3Z,6Z,9S,10R)-9,10-epoxy-3,6-heneicosadiene (Z3, Z6-9S, 10R-epoxy-21Hy) and (3Z,6Z,9S,10R)-9,10-epoxy-1,3,6-heneicosatriene (1, Z3, Z6-9S, 10R-epoxy-21Hy), which are produced by female *H. cunea* (*Tóth et al., 1989*). There are two major groups of moth sex pheromones: Type I pheromones and Type II pheromones (*Tóth et al., 1989*; *Millar, 2000*; *Ando, Inomata & Yamamoto, 2004*). Type I pheromones mostly contain $C_{10}$–$C_{18}$ unsaturated hydrocarbons and a terminal functional group (>75% moth species). Type II pheromones lack a terminal functional group and contain $C_{17}$–$C_{23}$ unsaturated hydrocarbons and epoxy derivatives (*Millar, 2000*; *Ando, Inomata & Yamamoto, 2004*). Compared to many well-studied Type-I sex pheromone-producing moth species, the molecular mechanisms of olfaction in the Type-II sex pheromone-producing *H. cunea* are poorly understood. In the current study, a total of 10 CXE genes were identified based on our previous *H. cunea* antennal transcriptomic data (*Zhang et al., 2016*). To understand the potential physiological roles of these HcunCXEs, we constructed a phylogenetic tree to evaluate the relationship of HcunCXEs with other insects' CXEs, and used reverse transcription-quantitative PCR (RT-qPCR) and reverse transcription PCR (RT-PCR) to investigate the expression of these genes. We found that HcunCXEs displayed either antennae- or leg/wing-biased expression. The differential expression pattern of HcunCXEs suggests a potential function in degrading pesticides and/or other xenobiotics.

## MATERIALS AND METHODS

### Insect rearing and tissue collection

*H. cunea* pupae were collected from a first-generation population at Baimao Town, Jiujiang District, Wuhu City, Anhui province. Insect cages were used for rearing *H. cunea* pupae at 25 °C, 70–80% RH and 14L:10D hour photoperiod. After eclosion, adults were provided with 1% honey water. In the fourth hour of the second dark period, antennae, thoraxes, abdomens, legs, and wings of virgin males and females were dissected under the microscope and pooled by sex and body part. Male and female pupae and fourth instar larvae were also sampled. Five samples were taken for each body part with the exception of antennae, of which 30 pairs were collected by pulling out from the base of the antennae with tweezers. Dissected body parts or whole-body samples were flash frozen in liquid nitrogen and stored at −80 °C until use.

### Gene annotation

The *H. cunea* antennal transcriptome (PRJNA605323) (*Zhang et al., 2016*) was used as a reference sequence for mapping clean reads for each tested sample. Gene annotation was carried out using Nr (NCBI non-redundant protein sequences), Nt (NCBI nucleotide), Pfam (Protein family), KOG/COG (Clusters of Orthologous Groups of proteins/enKaryotic Ortholog Groups), Swiss-Prot (A manually annotated and reviewed protein sequence database), KEGG (Kyoto Encyclopedia of Genes and Genomes) and GO (Gene Ontology) databases (Figs. S1–S4). Based on the results of gene annotation and BLAST comparison, the candidate genes of HcunCXE were determined and named according to the identification order from the antennal transcriptomic data.

### Homologous search and sequencing analysis of CXE genes in *H. cunea*

The *H. cunea* CXE genes were identified according to the BLAST results on NCBI. The Open Reading Frame finder (OFR Finder) (https://www.ncbi.nlm.nih.gov/orffinder/) was used to search for the open reading frame of these CXE genes. An ExPASy tool (http://web.expasy.org/compute_pi/) (*Petersen et al., 2011*) was used to calculate their theoretical isoelectric points (pI) and molecular weights (MW) of the full-length HcunCXEs gene candidates, and SignalP-5.0 (https://services.healthtech.dtu.dk/service.php?SignalP) was used to predict signal peptides of the CXE genes (*Petersen et al., 2011*).

### Phylogenetic analysis of CXE genes in *H. cunea*

Genes related to the CXEs of *H. cunea* and other reported insects (*Seasamia inferens, Spodoptera littoralis, Spodoptera exigua, Cnaphalocrocis medinalis, Bombyx mori, Drosophila melanogaster, Tribolium castaneum, Mamestra brassicae* and *Antheraea polyphemus*) were subjected to multi-sequence alignment with MAFFT (*Wong, Suchard & Huelsenbeck, 2008*). Amino acid sequences were automatically aligned by the MAFFT program version 7 (http://mafft.cbrc.jp/alignment/software/algorithms/algorithms.html), using L-INS-i strategy (*Katoh & Standley, 2013*). The phylogenetic tree was constructed using MEGA-X (*Tamura et al., 2011*) and maximum likelihood method (1000 bootstrap

repetitions) for systematic evolution analysis. The adopted model was LG-G+I, and all sites were used for Gap/Missing Data Treatment. Lastly, the phylogenetic tree was edited on the website iTOL (https://itol.embl.de/). The genes of insect ODEs required for the phylogenetic tree are shown in Table S1.

## RNA extraction and synthesis of the first-strand cDNA

The sampled body tissues were ground using a Tissue-Tearor which rapidly homogenized the samples in DEPC-treated sterile water. TRIzol reagent (Invitrogen, USA) was used for extraction and purification of total RNA from each sample according to the manufacturer's instructions. The degradation and contamination of RNA was monitored on 1% agarose gels, and purity was checked using a NanoPhotometer® spectrophotometer (IMPLEN, CA, USA). First-stranded cDNA templates were synthesized using 1 μg of RNA template with the PrimeScript™ RT reagent Kit according the manufacturer's instructions (TaKaRa, Japan).

## RT-qPCR and RT-PCR analysis

Expression profiles of the identified *H. cunea* CXE genes in different body parts of adults and two other life stages were analyzed. Tissues included antenna of 30 adults of each sex, legs of 5 adults of each sex, wings of 5 adults of each sex, thoraxes and abdomens of 5 adults of each sex, 5 whole pupae of each sex and 5 larvae (fourth instar).

The RT-qPCR and RT-PCR assays were employed for production of multiple copies of DNA. RT-qPCR reaction was conducted in a 25 μL reaction mixture system containing 12.5 μL of SYBR® Premix Ex Taq II (Tli RNaseH Plus) (TaKaRa, Japan), 1 μL of each primer, 2 μL of sample cDNA, and 8.5 μL of sterilized $H_2O$.

The RT-qPCR cycles were set at 95 °C for 30 s, followed by 40 cycles at 95 °C for 5 s, 60 °C for 30 s. Each experiment was carried out in a CFX96 real-time PCR detection instrument (Bio-rad, USA) using 8-strip PCR tubes (Bio-rad, USA). The reaction data were recorded, and the dissolution curves were appended. Both Elongation factor-1 alpha (EF1-a) and glyceraldehyde-3-phosphate dehydrogenase (GAPDH) were used as internal reference. Three biological replicates were performed, and the reproducibility confirmation of each RT-qPCR reaction was replicated three times for each sample (Table S2) (*Xu et al., 2018*).

The variability of each gene expression in different body tissues was tested by using the Q-Gene method (*Muller et al., 2002*; *Simon, 2003*). The relative expression of mRNA of each gene (mean ± SD) was analyzed using one-way ANOVA (SPSS22.0 for Windows, IBM, USA), followed by LSD and Duncan's tests at $\alpha = 0.05$. GraphPad Prism v5.0 Software (GraphPad Software Inc, CA, USA) was used for graphical plotting/mapping.

RT-PCR analysis was performed as follows: 94 °C for 2 min of initiation, and 29 cycles of 94 °C for 30 s, 52 °C for 30 s, 72 °C for 15 s, and 2 min at 72 °C for final extension. Elongation factor-1 alpha (EF1-a) of *H. cunea* was used as an internal reference. In addition, instead of template cDNA, RNase-free water was used as the blank control. The reaction mixture contained 12.5 μL of 2x Ex Taq MasterMix (CWBIO, China), 1 μL of each primer, 1 μL of sample cDNA, and $H_2O$ to bring the total to 25 μL. A 10 μL aliquot of each reaction product was used for gel electrophoresis. The RT-PCR primer sequences of CXE genes in *H. cunea* are listed in Table S3.

**Table 1** Gene name, information of open reading frame and Blastx match of the 10 putative HcunCXEs identified in this study.

| Gene name | ORF length (bp) | Complete ORF | FPKM value | Best Blastx Match | | | |
|---|---|---|---|---|---|---|---|
| | | | | Species | Acc.number | *E* -value | Identity (%) |
| HcunCXE1 | 1668 | YES | 4.9 | *S. inferens* | AII21990.1 | 0.0 | 73 |
| HcunCXE2 | 777 | NO | 3.77 | *S. inferens* | AII21980.1 | 3e−135 | 73 |
| HcunCXE3 | 375 | YES | 3.26 | *S. inferens* | AII21980.1 | 2e−105 | 60 |
| HcunCXE4 | 1389 | YES | 61.01 | *S. inferens* | AII21984.1 | 0.0 | 59 |
| HcunCXE5 | 1593 | YES | 143.14 | *S. inferens* | AII21984.1 | 0.0 | 62 |
| HcunCXE6 | 1161 | NO | 17.04 | *S. inferens* | AII21984.1 | 4e−174 | 62 |
| HcunCXE7 | 1677 | YES | 13.18 | *S. inferens* | AII21987.1 | 0.0 | 75 |
| HcunCXE8 | 1608 | YES | 12.64 | *S. inferens* | AII21980.1 | 0.0 | 66 |
| HcunCXE9 | 1653 | YES | 6.13 | *S.inferens* | AII21978.1 | 0.0 | 71 |
| HcunCXE10 | 273 | NO | 21.32 | *S. inferens* | AII21984.1 | 8e−39 | 64 |

Notes.

ORF, open reading frame; *S. inferens*, *Sesamia inferens*.

**Table 2** Gene name and characteristics including molecular weight, isoelectric point and signal peptide of the 10 putative HcunCXEs with open reading frames.

| Gene Name | MW (Kda) | PI | SP |
|---|---|---|---|
| HcunCXE1 | 62.23 | 7.56 | NO |
| HcunCXE2 | 28.44 | 5.67 | NO |
| HcunCXE3 | 13.98 | 4.85 | NO |
| HcunCXE4 | 52.2 | 5.31 | NO |
| HcunCXE5 | 59.52 | 5.41 | NO |
| HcunCXE6 | 43.17 | 5.09 | NO |
| HcunCXE7 | 61.71 | 6.32 | 1-17 |
| HcunCXE8 | 60.68 | 5.75 | NO |
| HcunCXE9 | 62.18 | 8 | 1-16 |
| HcunCXE10 | 10.52 | 8.89 | NO |

Notes.

SP, signal peptide; pI, isoelectric point; MW, Molecular weight.

## RESULTS

### Identification of CXE genes from *H. cunea*

Based on a comparative analysis of the *H. cunea* antennal transcriptome using BLASTX databases (*Zhang et al., 2016*), a total of 10 HcunCXE genes were identified. BLASTX comparison showed that these 10 HcunCXE genes have high homology with CXE genes of *S. inferens*. Six HcunCXEs (HcunCXE1, HcunCXE3-5 and HcunCXE7-8) had complete ORFs (Table 1). The molecular weights of these HcunCXEs ranged from 10.52 to 62.23 kDa (Table 2). Only HcunCXE7 and HcunCXE9 have predicted signal peptide sites (Table 2).

## Phylogenetic analysis of *H. cunea* CXEs

To evaluate the relationship of HcunCXEs with other insects' CXEs, a phylogenetic tree was constructed (Fig. 1). The HcunCXEs genes could be divided into two subclasses: extracellular gene subclass (generally secreted enzymes, substrates include hormone and pheromones) and generally intracellular enzymes, dietary metabolism/ detoxification functions (Fig. 1). Three HcunCXEs (HcunCXE1, 7 and 9) were clustered in the generally secreted enzymes subclass. The other 7 HcunCXEs including HcunCXE2-6, HcunCXE8 and HcunCXE10 fell into the intracellular gene subclass. In addition, the clade of intracellular gene subclass formed by HcunCXEs was most closely related to those formed by *S. inferens, C. medinalis, S. exigua* and *S. littoralis* CXEs. Sequence alignments showed that the amino acid identities of HcunCXE1 and SinfCXE18, HcunCXE9 and SinfCXE1, HcunCXE7 and SinfCXE13, HcunCXE7 and CmedCXE5 were 73.9%, 71.3%, 74.6% and 65%, respectively (Fig. S5). These results suggest that the intracellular CXEs in *H. cunea* shared a more recent common ancestor with the CXEs in *S. inferens, C. medinalis, S. exigua* and *S. littoralis* than with the CXEs in other insect species.

## Tissue distribution of HcunCXEs

We next examined the expression of HcunCXE genes in adult female and male antennae, legs and wings using RT-qPCR with primers specific for each of the 10 HcunCXEs genes (Table S2). All HcunCXEs were expressed in the antennae (Fig. 2 and Fig. S6). Among which, three HcunCXEs (HcunCXE4, 5, 8) were highly expressed in the antennae (Figs. S6C and Figs. S6D). Two HcunCXEs (HcunCXE1 and 3) were female-biased (Figs. 2A and 2C) and two HcunCXEs (HcunCXE 9 and 10) were male-biased (Figs. 2I and 2J); although the sex-biased expression is not statistically significant, there is a clear numerical difference between expression level in the sexes. These results indicate that the most abundant CXE genes in the antenna are not extracellular CXEs that likely participate in volatile odorant degradation. The most abundant CXEs are likely involved in primary metabolic activities and it would thus seem logical that their expression is much higher than for the other specialized CXEs in the antenna. The other HcunCXEs, however, were equally expressed in both sexes. Comparing expression across tissues, five HcunCXEs (2, 3, 5, 7 and 8) were highly expressed in the legs and wings (Figs. S6A and S6B). Expression of HcunCXE2 and HcunCXE7 was higher in the legs or wings than that in the antennae (Figs. 2B and 2G).

To investigate whether these HcunCXEs are also expressed in the other body parts or life stages, a RT-PCR experiment was carried out using total RNA samples taken from *H. cunea* adults and other life stages (pupae and larvae). Gel electrophoresis bands were generated from HcunCXE2 products from the adult thoraxes and abdomens (Fig. 3). In addition, faint/light bands of HcunCXE7 and HcunCXE8 were detected in both thoraxes and abdomens, as well as the pupae. Interestingly, nine out of 10 HcunCXEs (HcunCXE1-5 and 7–10) were also detected in the larvae, indicating that HcunCXEs are widely expressed in the larval stage.

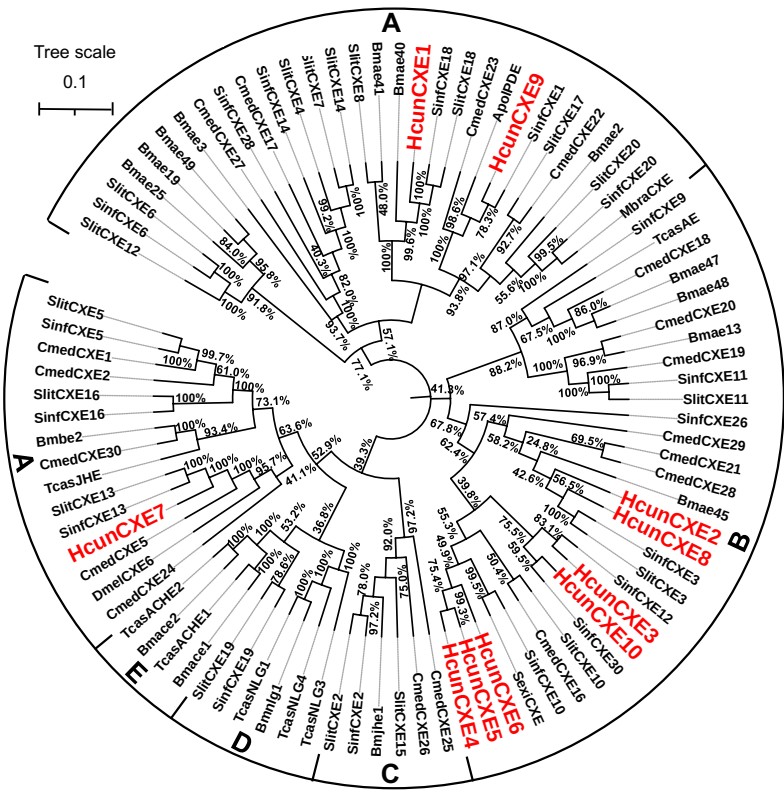

**Figure 1 Molecular phylogeny comparing HcunCXEs with CXEs from other insect species.** 10 CEXs (HcunCXE1-10) from *H. cunea* (Hcun) and CXEs from *S. exigua* (Sexi), *C. medinalis* (Cmed), *B. mori* (Bmor), *D. melanogaster* (Dmel), *T. castaneum* (Tcas), *S. inferens* (Sinf), *S. littoralis* (Slit) were used to construct the phylogenetic tree. The phylogenetic tree was aligned by MAFFT, and constructed by MEGA-X using maximum likelihood method. The adopted model is LG-G+I, and the model value is shown in Table S4. The Bootstrap value of this tree is 1,000, which is to integrate the branch length tree with the Bootstrap value tree and then beautify it. (A) Extracellular gene subclass (Generally secreted enzymes, substrates include hormone and pheromones); (B) generally intracellular enzymes, dietary metabolism/detoxification functions; (C) juvenile hormone esterase (JHE); (D) nerouligins; (E) acetylcholinesterases (AChE).

## DISCUSSION

In the current study, 10 putative CXE genes were identified based on our previous *H. cunea* antennal transcriptomic data (*Zhang et al., 2016*). All 10 *H. cunea* CXE genes showed high homology to the CXE genes identified in *S. inferens* (identity ≥59%, Fig. 1 and Table 1). We speculated that some of these *H. cunea* CXE genes mainly degrade sex pheromone components and host plant volatiles. Unlike many well-studied Type-I sex pheromone-producing lepidopteran insects (>75% moth species), the *H. cunea* sex pheromone is comprised of Type II pheromone components (*Ando, Inomata & Yamamoto, 2004*). At present, most of the published moth ODEs are from the Type I sex pheromone producing lepidopterans; thus, our study represents the first report of ODE genes from a Type II sex pheromone-producing moth species. *H. cunea* is an extremely polyphagous species with high fecundity (several hundred eggs/female) and dispersal capacity. *H. cunea* larvae

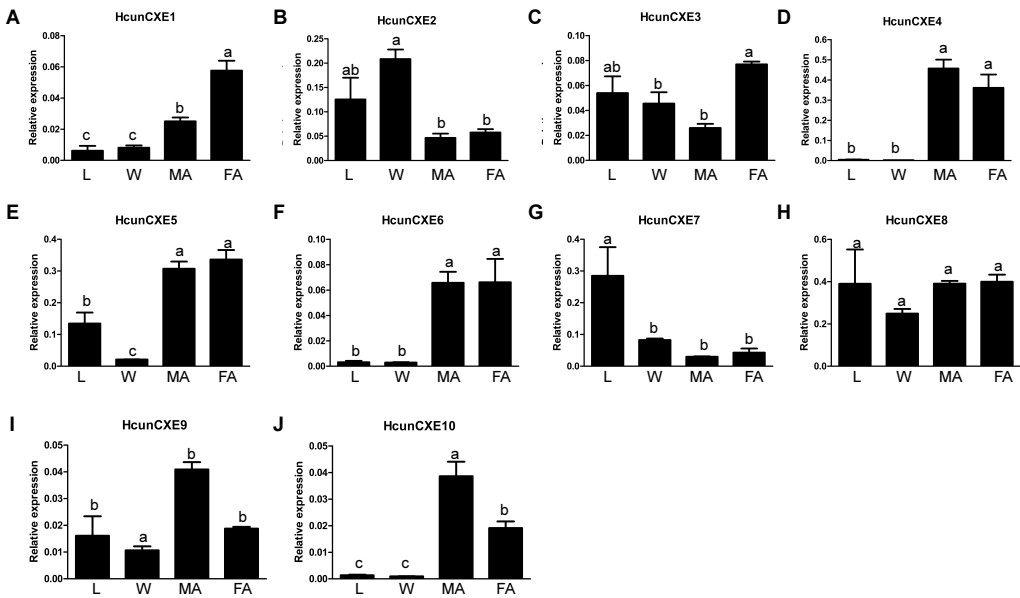

**Figure 2** **Relative mRNA expression of *HcunCXEs* in *H. cunea* tissues.** (A–J) HcunCXEs (HcunCXE1, 2, 3, 4, 5, 6, 7, 8, 9 and 10). FA, female antennae; MA, male antennae; L, legs; W, wings. The relative mRNA levels were normalized to those of the *EF1-a* gene and analyzed using the Q-gene method. All values are shown as the mean ± SEM. The data were analyzed by the least significant difference (LSD) test after one-way analysis of variance (ANOVA). Different letters indicate significant differences between means ($P < 0.05$).

are generalists, capable of feeding on over 170 species of host plants, including many broad-leaved tree species. To cope with such diverse host plant species, this moth must have developed a series of olfactory receptor neurons to recognize diverse plant volatiles (*Zhang et al., 2016*). The number ($n = 10$) of CXE genes we identified from *H. cunea* was lower than those of other reported lepidopterans species: 19 in *Chilo suppressalis*, 35 in the tea geometrid *Ectropis obliqua* Prout and 76 in *B. mori* (*Yu et al., 2009*; *Liu et al., 2015*; *Sun et al., 2017*). These results suggest that *H. cunea* does not seem to require more CXEs, since the other ODEs including CYP, AD, AOX and GST are likely involved in odorant degradation in olfactory processes. On the other hand, the difference in number of CXEs in various species might result from differences in sample preparation and sequencing method/depth. In addition, the ecological/evolutionary differences across species may also be a reason. Insects have to adapt to their external environment; different environments lead to the formation of different physiological and behavioral characteristics.

The phylogenetic tree analysis showed that HcunCXE1, 7 and 9 belong to the extracellular gene subclass, including the secretory enzymes that likely act on hormones and pheromones (Fig. 1). The remaining 7 CXE genes fell into the intracellular gene subclass (Fig. 1), including intracellular enzymes that mostly play roles in dietary metabolism and detoxification. *Chertemps et al. (2012)* demonstrated that an extracellular CXE of *D. melanogaster*, esterase-6 (EST-6), is responsible in or related to the sensory physiological and behavioral responses to its pheromone. A subsequent study found that EST-6 was

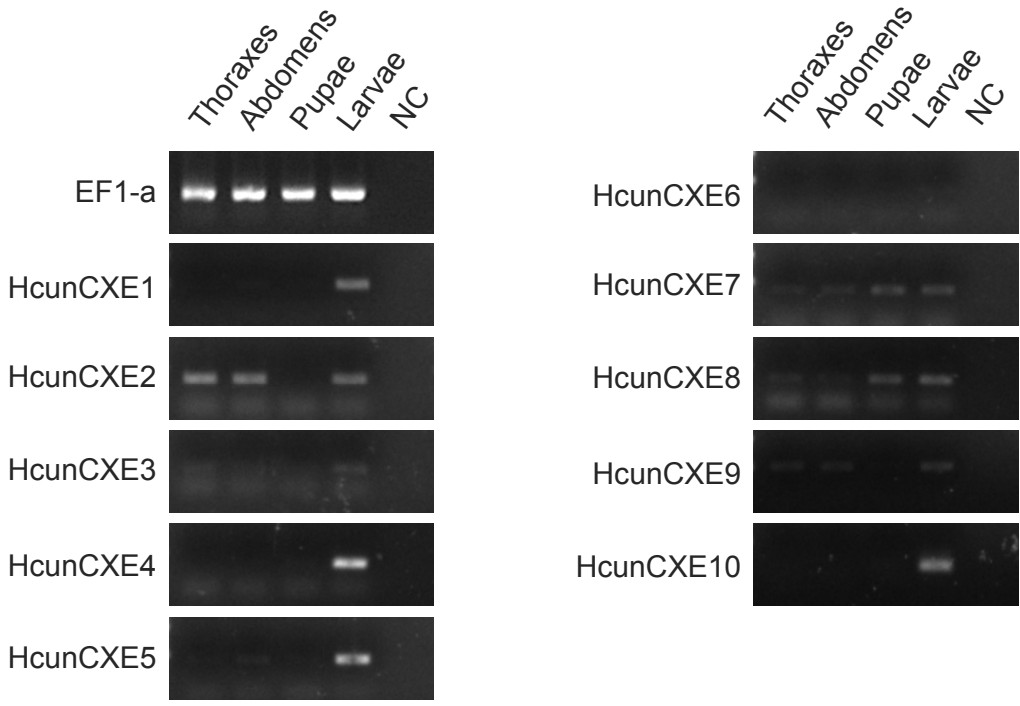

**Figure 3** **RT-PCR analysis of HcunCXEs gene expression in tissues taken from *H. cunea* adults and other life stages.** EF1-a was used as an internal control; NC, negative control with no template in the reaction.

able to degrade various volatile esters in vitro and function as expected for an ODE which plays a role in the response of the flies to esters (*Chertemps et al., 2012*). Thus, these *H. cunea* CXE genes (HcunCXE2, 3, 4, 5, 6, 8 and 10) may also affect the mating and courtship competitions in *H. cunea* through degradation of some ester kairomones or plant allelochemicals. On the other hand, based on the omnivorous nature of *H. cunea* and its species-specific sex pheromone, these CXE genes may be a unique category of *H. cunea* which degrade odor substances.

Antennal-specific or highly expressed esterases belong to the CXE type in the carboxy/cholinesterases (CCEs) family. The first ODE was identified form *A. polyphemus* (ApolSE) as an antenna-specific esterase, with a high ability to degrade the acetate component (*E6Z*11-16: AC) of its pheromone blend (*Vogt & Riddiford, 1981*). Since then, antennal-specific esterases have been cloned from *A. polyphemus* (*Ishida & Leal, 2002*) and *Mamestra brassicae* Linnaeus (*Maïbèche-Coisne et al., 2004*). Recent studies show that many insect CXEs are expressed specifically in antennae, and their major functions in olfactory process are to degrade odor molecules. Interestingly, the expression of some HcunCXEs in the legs and wings were found to be higher than those in the antennae (HcunCXE2, 3 and 7). The ten *H. cunea* CXEs genes we identified through the gene expression analysis had a low level of expression in different body tissues of *H. cunea* adults (Fig. 2 and Fig. S6). However, they were widely expressed in the larvae, which may be related to their extremely broad host plant range that needs more CXEs to degrade large amount of

carboxylic acid esters. Our quantitative PCR results (Fig. 2 and Fig. S6) indicated that some HcunCXEs genes were highly expressed in both male and female antennae. HcunCXE1 and HcunCXE9 belong to the same subclass as ApolPDE and MbraCXE (Fig. 1). Previous studies have shown that ApolPDE and MbraCXE function as pheromone degradation enzymes (*Maibeche-Coisne et al , 2004*; *Ishida & Leal, 2005*). These HcunCXEs are likely for degradation of sex-pheromones and/or plant volatiles both from hosts or non-hosts. However, the HcunCXEs genes that were highly expressed in the legs and wings might be related to the degradation of non-volatile substances for contact signals. In addition, a previous study of SexiCXE14 and SexiCXE15 (antennae-enriched carboxylesterase genes in *Spodoptera exigua*) showed that antenna bias expression plays a role in the degradation of volatile substances and sex pheromones in plants (*He et al., 2015*). However, the expression of SexiCXE11 was much higher in abdomen and wings, and its activity in hydrolyzing plant volatile substances was stronger than that in degrading ester sex pheromones (*He et al., 2019*). In the current study, HcunCXE1, 3, 4, 5, 6, 8, 9, and 10 showed antenna-biased expression, while the expression of HcunCXE2 and 7 in legs and wings was higher than that in antennae. These results suggested that HcunCXEs have different functions and may participate in the degradation of host plant volatiles and/or other xenobiotics.

CXEs play multiple key roles in the hydrolysis of carboxylic acids esters. CXEs also include some metabolic enzymes that are associated with insecticide resistance (*Li, Schuler & Berenbaum, 2007*). Many previous studies in insect CXEs focused on their functions in mediating insecticide resistance (*Hemingway & Karunaratne, 1998*; *Li, Schuler & Berenbaum, 2007*). In contrast, the mechanisms underlying degradation of plant allelochemicals are still unclear. It has been shown that phenolic glycosides can induce expression of *Papilio canadensis* CXEs (*Lindroth, 1989*). Moreover, in *Lymantria dispar*, the activities of CXEs were positively correlated with the larval survival, indicating that these esterases might be involved in the glycoside metabolism (*Lindroth, 1989*; *Lindroth & Weisbrod, 1991*). In the current study, nine out of 10 HcunCXEs were expressed in the larvae (Fig. 3), indicating that the activities of HcunCXEs may positively correlate with survival of *H. cunea* larvae. Although the gene expression of HcunCXEs in *H. cunea* midgut and some other tissues are still unknown, based on these previous findings, it is reasonable to speculate that HcunCXEs might also play multiple functions in *H. cunea* physiology and metabolism. In addition, a significant increase of CXE activity in the midgut of *S. litura* was observed during uptake of the plant glycoside rutin (*Ghumare, Mukherjee & Sharma, 1989*). The CXEs in *Sitobion avenae* have been suggested to participate in gramine detoxification (*Cai et al., 2009*). Quercetinrutin and 2-tridaconone were also found to induce the activities of CXEs in *Helicoverpa Armigera* (*Gao et al., 1998*; *Mu, Pei & Gao, 2006*). Understanding the specific function of HcunCXEs will require further analyses using in vitro and in vivo methods.

Little is known about *H. cunea* olfaction mechanisms at the molecular level, especially concerning how CXEs degrade various semiochemicals in its chemical communication system. Further research is needed to (1) understand the functions of antennal-specific CXEs in *H. cunea* via cloning, expression and purification of these CXEs and enzymatic kinetic analysis; (2) determine the locations/distributions of related CXEs by *in-situ*

hybridization; (3) evaluate the potential correlations between CXE transcription levels and their corresponding electrophysiological and behavioral responses by silencing CXEs via RNA interference (*Caplen, 2004*), and (4) ultimately discover the mode of action or functionality of CXEs in the olfactory signal conduction (signal inactivation).

## CONCLUSIONS

In summary, we identified 10 CXE genes in *H. cunea* by analyzing its antennal transcriptomic data. These HcunCXEs displayed an antennae-or leg/wing-biased expression. The ubiquitous expression of these HcunCXEs in different tissues and life stages suggest that they have multiple roles, i.e., degradation of odor molecules, metabolism and detoxification of dietary and environmental xenobiotics. Our findings provide a theoretical basis for further studies on the olfactory mechanism of *H. cunea* and offer some new insights into functions and evolutionary characteristics of CXEs in lepidopteran insects. From a practical point of view, these HcunCXEs might represent meaningful targets for developing behavioral interference control strategies against *H. cunea*.

## ACKNOWLEDGEMENTS

We would like to thank Dr. Jacob D. Wickham (Managing Editor, Integrative Zoology), Dr. Melissa Matthews and Dr. Hong Huat Hoh (OIST Graduate University, Japan) for editing the manuscript, Dr. Tianzi Gu and Zhenchen Wu for helpful suggestions.

### Funding

This project was funded by the National Key Research and Development Program (2018YFC1200400), the National Natural Science Foundation of China (31870640), the Natural Science Foundation of Anhui province, China (1508085SMC216), and the National Research Innovation Program for undergraduates Graduates (201710364019). The funders had no role in study design, data collection and analysis, decision to publish, or preparation of the manuscript.

### Grant Disclosures

The following grant information was disclosed by the authors:
National Key Research and Development Program:  2018YFC1200400.
National Natural Science Foundation of China:  31870640.
Natural Science Foundation of Anhui province, China: 1508085SMC216.
The National Research Innovation Program for undergraduates Graduates:  201710364019.

### Competing Interests

Qing-He Zhang is an employee of Sterling International, Inc., Spokane, WA.

## Author Contributions

- Jia Ye conceived and designed the experiments, performed the experiments, analyzed the data, prepared figures and/or tables, authored or reviewed drafts of the paper, and approved the final draft.
- Dingze Mang conceived and designed the experiments, analyzed the data, prepared figures and/or tables, authored or reviewed drafts of the paper, and approved the final draft.
- Ke Kang conceived and designed the experiments, performed the experiments, prepared figures and/or tables, and approved the final draft.
- Cheng Chen and Xiaoqing Zhang performed the experiments, prepared figures and/or tables, and approved the final draft.
- Yanping Tang performed the experiments, authored or reviewed drafts of the paper, and approved the final draft.
- Endang R. Purba, Liwen Song and Qing-He Zhang analyzed the data, authored or reviewed drafts of the paper, and approved the final draft.
- Longwa Zhang conceived and designed the experiments, analyzed the data, prepared figures and/or tables, authored or reviewed drafts of the paper, and approved the final draft.

## Data Availability

Raw measurements are available in the Supplementary Files.

## Supplemental Information

Supplemental information for this article can be found online at http://dx.doi.org/10.7717/peerj.10919#supplemental-information.

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
