# Peer review of "Putative carboxylesterase gene identification and their expression patterns in Hyphantria cunea (Drury)"

_PeerJ, doi:10.7717/peerj.10919_

## Round 0.1 · original submission · Major Revisions

Please pay close attention to detail when addressing all the Reviewers' comments. All Reviewers have provided detailed comments which I agree with. Also, please consider having the manuscript edited for English grammar.

Reviewer 1 ·

Basic reporting

o In terms of English quality it looks regular. Some words seem odd in this type of research field, i.e. “olfactorily” in the abstract (Line 25). Similarly, in Line 31 “our results indicated” should be substituted by “our results indicate”. Line 245 “Till now” is not proper. There are some typos: see line 111 (“ware”). I hence would recommend to send the manuscript for English revision to a native speaker researcher from our line of study.
o In line 62, I do not consider that the part of “as well as herbicide activation” is pertinent because it seems out of place in this kind of manuscript. Furthermore the citations on this part refer much more to insecticide resistance, which although there is a body of publications about herbicides, could lead the reader to mistrust if they meant herbicide or insecticide.
o Line 69, should include, of course, Drosophila melanogaster, they can use the reference of Chertemps 2012 or the one from 2015.
o At line 80 they should add as well the main findings on Chertemps et al., 2015.
o Line 95, I think is pertinent to write a couple of lines about the identity of the pheromone components of the pheromone blend of H. cunea.
o At line 97, the citation is not italics as all the rest.
o Re-write the sentences between 125-128 because as it is right now don’t make any sense.
o Line 196, leave it until “subclass” the rest is rather obvious.
o Figure 1 legend needs major corrections: first try to state under the figure the type of tree (Maximum likelihood), the model used (see my comment on substitution model) and the bootstrap, do not only say to go back to the methodology for details. Indicate that the numbers are also placed on the tree to indicate bootstrap support.
o In the phylogeny analysis part in results it would be useful to state the percentages of identity between HcunCXEs and Dmel, Sinf and Slit at least for the most closely related.
o In line 221, it must be acknowledged that the sex-biased expression is not statistically significant but there is a clear numerical difference between expression in the sexes.
o Lines 235-239 are not really relevant in here. If they want to introduce the importance of ODEs they should use the explanation given by Leal (2013) on how ODEs must act fast for plume determination.
o Lines 240-241: how much homology? What percentage of identity? This statement is not obvious just by looking at the tree.
o Line 244: this is a good time to mention the pheromone components.
o Line 262: but it could also act in a broader function. They should see paper from 2015 and related to their results.
o Line 287: correct this scientific name! the species name is ALWAYS in lowercase letters
o It is important that they share the protein sequences as raw data. This is also important for reviewers.

Experimental design

I think it does comply with the aims and scope, the research question is well-defined and meaningful. However, I have some problems with the methodology used and described:

1.My main question is how did they find the 10 ODEs? They mention that they use the information from their 2016 transcriptome, but I want to know how they performed the extraction of these candidate genes. As it is right now, they just state that used the transcriptome to check for transcript abundance but it does not make clear how they actually annotated candidate ODEs. I guess they just ran some kind of BLAST to their previously determined transcriptome, but it is important to state it, along with the parameters to consider as reliable predictions (e-value, % identity, hits, etc). I would add a whole new section “Gene annotation” or something similar. Furthermore, I want to know what method did they use to name the new genes? Based on what? It does not look like they based it on transcript abundance or similarity to other species sequences (S. inferens for example), so please explain.
2. I have a problem with the methodology followed to construct the phylogenetic tree. The first thing I recommend is to go ahead and state in lines 131 and 132 all the species from which ODE sequences were used to make the comparison. I also personally prefer MAFFT for multiple sequences alignment but I guess Clustal is fine. It is however necessary that they perform a best substitution model analysis to make the phylogenetic tree, after all, choosing the proper evolutionary model can affect the topology of the tree, branch length estimation the statistical support values. They also will need to state in the figure or elsewhere which is the selected gene evolution model. Considering that there are several newer versions of MEGA I would also recommend to use newer versions if possible.
3. I would recommend to state how they normalize the data of relative RNA expression in the methodology.

Validity of the findings

No comment

Additional comments

Your article fills up an important gap since ODEs have not been as extensively investigated as other chemoreceptor proteins for example. Nevertheless, I strongly recommend to implement a series of small changes to make it more scientifically sound. First and most importantly to check again for language mistakes and some others changes to increase the reliability of your findings.

·

Basic reporting

a. Language: There is a high number of spelling/grammatical errors throughout the entire manuscript including Tables and Figures. There are also some typographical errors such as extra spaces. (eg. Line 102) Several statements must be improved for clarity and overall understanding of the text (Please refer to the highlighted text in manuscript pdf and comments below). The following lines represent examples for which language could be improved:
Line 53-54: This statement can be improved for clarity and also requires a reference.
Line 55-59: This statement can be improved for clarity and scientific accuracy. These families of enzymes are distinctly different based on their protein structure and function; however, they have been shown to catalytically interact with odor molecules of the same type and structure; ie. it is currently believed that multiple ODEs (belonging to the different enzyme families) may participate in degradation and clearing of the same type of odor molecule (See Steiner et al. 2019)
Line 65-67: This statement can be improved for clarity and grammatical correctness.
Line 100-102: This statement can be improved for clarity. Is it the differential expression pattern that suggests the function? Is the expression of the gene enough to suggest a function?
Line 185: It is not clear why the CXE genes were specifically compared to S. inferens.
Line 217-218: RT-qPCR can’t be used to probe the physiological function of HcunCXEs, it can be used to explore differential expression patterns of the transcripts of interests between different tissue types and sexes. The statement should be reworked for clarity of what is being investigated. Inferring physiological function from transcript expression level is speculative at best and must be clearly identified as such.
Line 251-254: The OR landscape of H. cunea is known from the 2016 transcriptomic analysis and should, therefore, be referenced here. This statement should also be clarified for the readers. The number of CXEs identified is not that surprising given our current understanding of CXE’s function in clearing volatile molecules from the sensillar lymph. Please refer to Steiner et al 2019, Durand 2010a,b Durand 2016.
Line 282-294: How are the H.cunea results relevant or related to the content of the discussion here? The authors could improve this paragraph by providing relevant explanation to the results from H. cunea.
b. References: Several statements are missing relevant references. (Example: Line 53-53; Line 64-65) vi. The references list lacks in consistency and accuracy of the information provided. Some of the journal titles are in italics and some are not, some are represented as abbreviations and some are not. The titles are also not consistently capitalized. The references also contain a number of spelling and typographical errors.
c. Professional article structure, figures, tables: The structure of the paper is well organized and logical. The figures contain spelling errors and several of the figures could be represented better. Raw data shared is adequate.
Fig 1: The figure could be improved by having the Neurosignalling (red) and Generally intracellular enzymes (green) aligned parallel with the page and the two labels in blue. The world generally can likely be omitted for clarity. The figure description should be able to stand alone without having to refer to materials and methods.
Fig 2: The figure could be improved by introducing a and b for the distinct portions of the figure to avoid an ambiguous statement such as “the upper parts listed.” The description which follows could be reworked for better clarity. It is not clear what 8 most common motif-patterns are. The word Homologous in the figure is spelled incorrectly.
Fig 3: The y-axis in this figure for all of the graphs shown contains a spelling error. The clarity of the figure could be improved by combining results for all CXEs for one tissue type on one graph to immediately be able to know which of the genes are abundant in the given tissue type.
Fig 4: This figure would benefit from having the tissues involved in RT-qPCR included for comparison purposes. This would provide a link for the two experiments and further shed light on the relative expression of the CXE genes in the different tissue types.
Tables 1 & 2: the descriptions should include an explanation for any abbreviations used in the table so that the tables can stand alone without having to refer to the text of the manuscript.
d. The descriptive work with H.cunea CXE sequences is relevant to the questions regarding signal deactivation in odorant processing.

Experimental design

a. The research presented falls within the aims and scope of the journal but many issues should be addressed before the paper can be published.
b. The research question is well defined, relevant & meaningful
c. No comment
d. Methods are not described with sufficient detail & information to replicate the experiments.
Line 107-109: The description of insect material collection, rearing, and tissue collection should be improved for clarity. It is not clear what the authors mean by 1st generation population and if the pupae collected at the site listed were reared to adulthood. Were the tissues collected from the 1st generation adults or several generations? There is no mention of diet or reference to previously established rearing protocol. Were the pupae and larvae used for tissue collections from the materials collected at the Baimao Town site or as offspring from the 1st generation adults?
Line 116-119: The software package version as well as parameters used in RSEM transcript abundance quantification are missing from the description for future reproducibility.
Line 123-125: Software version, reference, or web address as well as relevant parameters such as the minimum transcript length specified for the ORF prediction should be listed.
Line 133-134: MEGA5.0 parameters for the neighbor-joining method should be provided. What program was used to generate the final graphical tree representation and was the final edited?
Line 109-111 and 145-149: How many insects (ie. number of males and females) were the tissues extracted from? The authors could also describe the excision points for each tissue type for clarity and reader understanding, ie. what was the excision point for the antennal tissue? The TRIzol extraction protocol was performed according to the manufacturer’s protocol or was it modified in any way? It would be helpful if the overall description for the RNA extraction and cDNA synthesis provided more detail as there are several ways the purified RNA can be treated before cDNA synthesis.
Line 154-156: It would be helpful if authors could specify whether whole pupae and larvae were used in tissue collection as well as the number of insects used.

Validity of the findings

a. No comment.
b. The authors state that this study is performed based on previous trancriptomic work from 2016. Search in NCBI databases reveals two transcriptomes PRJNA513855 and PRJNA428142. The authors of this study should specify which transcriptome was used in this study using the proper accession number rather than SUB6944247. The raw data provided is adequate.
c. The results and discussion sections could be improved by emphasizing how the results from H. cunea correspond to previous findings in other insect species. Several statements lack precision and clarity (Please refer to Section 1 for examples) It would be helpful if the authors could reiterate in the conclusion which of the 10 CXEs had antennal expression and based on the results taken together were most likely to be implicated in signal deactivation at the level of antenna.
d. The authors do make several speculative statements that require clarification and identification as speculative.

Additional comments

1. The importance/relevance of conserved motifs in CXE protein sequences is not mentioned in the introduction. Thus, when it is mentioned in materials and methods as well as results it is difficult for the reader to understand why this analysis is being done in the first place. Perhaps a brief mention of the CXE motif structure characteristic to the different protein subclasses might help clarify the usefulness of this information.
2. The authors mention RSEM transcript abundance estimation in the materials and methods however, the results from this analysis are not used in the results and discussion section. If this analysis is not relevant it should be removed from the Materials and Methods section for clarity. I note that the FPKM values are reported in Table 1 but these results are not addressed in the text of the results and discussion. How do these results compare to quantitative PCR? How are they relevant to the analysis overall?
3. Consistency in writing: in some instances, the authors provide the version of the program or the manufacturer of the reagent but in many cases, they do not.

·

Basic reporting

There is a lack of clarity of the reporting of the materials and methods that preclude full analysis of some of the data being presented. This must be rectified before the manuscript can be adequately reviewed and assessed.

In general the language of the text is satisfactory. However, there are sufficient many grammatical errors that I recommend the authors ensure the manuscript is edited for language prior to re-submission.

Furthermore there are several minor errors in the presentation of the text and figures that indicate that the authors need to take greater care in attention to detail and these errors must be corrected before re-submitting. Specific examples are given below in the general comments to the authors.

Experimental design

This research study fits within the aims and scope of the journal.

Methods are not described with sufficient detail and information to replicate. Critical information is missing. Specific comments are provided in the general comments for the authors section below.

Validity of the findings

Underlying data has been provided, however, it is not clear in the supplemental data what the underlying data represents. Clarification is required. Specific comments presented below in the general comments to the authors.

It is difficult to assess the validity of the RT-qPCR data because MIQE guidelines for reporting aPCR studies have not been adhered to. Critical information is missing in the materials and methods that render it difficult to accurately assess the data.

Additional comments

Abstract.

Line 29. “previous H. cunea transcriptomic data”

The authors should clearly indicate here the source of the data. Which tissue. Later on in the article, it is mentioned that this is antennal transcriptome, but that should be made clear to the reader in the abstract.

Introduction

Line 53-54. “which may lead to disorders of the nervous system”.

This is vague and not clear what the authors mean. Greater specificity is required. For example, it could be said “which may lead to poor spatio-temporal resolution of the odor signal, and pose fatal hazards to the insects”

Line 66. Change “so as to” to “and”.

Line 97. “transcriptomic data”.

Again here, indicate that this is data from antennal transcriptome.

Line 99. “reverse transcription-quantitative PCR”.

Typically RT-qPCR refers to Real-Time quantitative PCR, not reverse transcription. Also abbreviations should be indicated in parenthesis foer both this and reverse transcription PCR, which follows.

Materials and Methods

Line 108. “Several body parts/tissues…”

It is essential to report how many insects were used for each tissue type. Also, how many biological samples were collected. This, especially, is important to know, with reference to the RT-qPCR study.

Line 131. “other reported insects were subjected”.

Which other insects. It is necessary in the Materials and Methods section to clarify which insects and motivate the choice of species, and specify where the sequences were obtained from (for example, from NCBI, or from directly from specific publications).

Line 131-132. “were subjected to multi-sequence alignment on Clustal…”
Line 133. “constructed using MEGA5.0…”

For Clustal, Mega, and all other bioinformatics software, it is necessary to indicated exactly which parameters were set before running the program to facilitate repeatability within the scientific community.

Line 138-139. “A total of 44 CXEs from…were used for identification”

Why were CXEs from these species chosen? Walker et al., 2019, BMC Genomics identified a larger repertoire of antennal expressed CXE genes in Spodoptera littoralis, for example. Using that as a reference set may provide more robust data.

Line 153. RT-qPCR and RT-PCR analysis section

How the RT-qPCR study, how many biological replicates were performed. From line 164-165, it is indicated that each reaction was replicated three times, but it is not clear if this is referring to technical replicates, or biological replicates. For this kind of study it is important to be clear in reporting.

Additionally, the authors need to consider MIQE guidelines for reporting on RT-qPCR studies. Bustin et al., 2009, Clinical Chemistry. The qPCR primers are mentioned, but there is no data shown on primer efficiency. In the materials and methods there is no mention of any control genes that have been used in the qPCR study. Control genes are essential. In the figure legend EF-1a is indicated, but in the supplemental data on the primers, both EF-1a and GAPDH primers are indicated. Was one of them used? Or both? According to MIQE standards, it is preferable to use more than one reference gene, but if only one is used, it must be shown that the reference gene shows stable expression across all tissue types.

Results

“shared a more recent common ancestor with the CXEs in D. melanogaster, S. inferens and S. litura”
Looking at Figure 1, it is not clear what basis the authors claim that the HcunCXEs shared more recent common ancestor with genes from D. melanogaster. The claim could surely be made for S. inferens and S. litura, but for every instance where homology is inferred for a D. melanogaster gene, the Hcun homologue is more closely related to genes from S. inferens and/or S. litura.

It also appears there is a taxa specific expansion of Hcun CXEs in the intra-cellular family. The authors should mention this more clearly in the results and discuss the implications of such expansions in the Discussion Section.

Line 210. “6-5-3-3-1-8-2-7-4”

Looking at Figure two, this looks like a typo. It seems there should only be one “3” here.

Discussion

Line 251-252. “Surprisingly, the number of CXE genes (n=10)…was much lower than those of other reported lepidopteran species:…”

Perhaps the authors should specify here if any of these other species utilize TypeII pheromones. Also should comment on why they think they may have found less ODEs….because of biological/ecological reasons or because of methodological reasons?
In looking at the Zhang et al., 2016 transcriptomic article on which this study is based, the numbers of ORs, GRs and IRs is indicative that the antennal transcriptome is fairly complete. However, to provide greater confidence in the completeness of the transcriptome, and thereby suggesting most of the CXEs in the transcriptome have been identified, I recommend the authors run BUSCO analysis on their transcriptome. gVolante server is a good source for this kind of analysis. https://gvolante.riken.jp/index.html

Line 291. “S. litur”

Should be “litura”?

Figure Legends and Figures

Line 518. “(Hyph)”

What is “Hyph”? Should be “Hcun”?

Figure 2. “Homolohous CXEs”

Should be Homologous. Also some of the Numbers of CXEs with each pattern is wrong. In the numbers column in the lower half of the figure. For the first row. (14) is indicated, but on the “Homologous CXEs column, it appears there is only 11. Likewise for the fourth row (16) is indicated, but it appears there is 15.

Supplementary Material 1 Raw Data
In the excel sheet it is not clear what the raw data numbers represent. This should be made clearer. What are those values indicating? Also the worksheet tab name is written in a Chinese script. For publication in an international journal, it should be written in English.

Supplementary Material 2 Raw Data

For the RT-PCR original gels, the columns need to be labelled so that the reader can understand what they are looking at. Which tissues were used as source of cDNA for amplification. This needs to be clear with labels on each column.

---

## Round 0.2 · Minor Revisions

Thank you for your revised manuscript. Please see below for my comments and those of the two Reviewers.

L25: change “olfactory important” to “important olfactory”

L26: change “Lepidopteran” to “lepidopteran”. Convention is to only capitalize orders when it is the true order name. Please update throughout the manuscript (L30 as well).

L50: I believe you can remove “different” from this sentence. “relevant” in this sentence implies that each insect species will make the choice that is best for them, and that they can both be similar and different.

L59: “glutathione-S-transferase” should be “glutathione S-transferase”. Please update throughout the manuscript.

L73: insert “have” between “CXEs” and “been”

L75: change “Antheraea polyphemus;” to “Antheraea polyphemus,”

L85: remove “the” before “plant volatiles”

L96: remove “the” before “local forests”

L105: Please correct the given reference, L106 as well.

L140: change “Blast” to “BLAST”

L156: Spelling. Change “Tribolium caastaneum” to “Tribolium castaneum”

L157 and 158: Please give the settings used for MAFFT and MEGA.

L160: Change “were” to “are”

L169: Change “MEME was done” to “MEME was run”

L169: Change “the width between the range of 6-10, and the number of motifs was below 8.” to “the width range was set to 6-10, and the number of motifs set to <8.”

L173: insert “a” before “Tissue-Tearor”

L174-176: change to “TRIzol reagent (Invitrogen, USA) was used for extraction and purification of total RNA from each sample according to the manufacturer’s instructions.”

L176: remove “product” and change “were” to “was”

L178: change “RNA templates” to “RNA template”

L179: change “manufacturer instructions” to “manufacturer’s instructions”

L187: insert “The” before “RT-qPCR”

L197: insert “the” before “Q-Gene method”

L198: change “expressions” to “expression”

L199: change “were” to “was”

L200-201: “The MIQE….” Please clarify and make into a complete sentence.

L201-202: Change “Graphical plotting/mapping was done by GraphPad Prism v5.0 Software (GraphPad Software Inc, CA, USA).” to “GraphPad Prism v5.0 Software (GraphPad Software Inc, CA, USA) was used for graphical plotting/mapping.”

L209-210: change sentence to “A 10 ul aliquot of each reaction product was used for gel electrophoresis.”

L215: change “Blastx” to “BLASTX” and throughout the manuscript.

L217-218: remove “As shown in Table 1” and place “(Table 1)” at the end of the sentence. Start the sentence with “Six HcunCXEs…”

L219: Remove “According to the prediction of the web server (Table 2)”. Start the sentence with “The molecular weights…” and place “(Table 2)” at the end of the sentence.

L220: Remove “The signal peptide predictions showed that” and start with “Only …”

L230: remove “found”

L238: The motif pattern analysis is not discussed at all in the Discussion. I also don’t find it very informative. What is known about these motifs? How does this support the conclusions you are drawing form this study? Please expand on it in your Discussion or consider removing the analysis. Also, in Figure 2, why aren’t the motifs presented from 1-8? Why is it presented as 8, 7, 6? Please correct this ordering.

L280-281: Consider changing “with a great fecundity (several hundred eggs/female) and a quick dispersal capacity.” to “with high fecundity (several hundred eggs/female) and dispersal capacity.”

L288-289: The acronyms for these genes were previously given, just use the acronyms in this sentence.

L301: Change “(Est-6)” to “(EST-6)”

L340-341: Reference for this work?

L344: Change “were found to express in” to “were expressed in”

L348” remove “the”

References: Please make sure that the formatting of all your references abides by PeerJ standards and requirements.

Please address all the comments by the Reviewers. Reviewer 2 has provided an Annotated PDF with additional comments.

I look forward to receiving your revised manuscript.

Reviewer 1 ·

Basic reporting

Looks much better now. However, there are pending revisions to be made, typos and statements to be changed for clarity. These are the ones I noticed:

Introduction
Please change "The molecule-bound odorant receptors (ORs) then convert the chemical signals into electrical signal that is... " by "The molecule-bound odorant receptors (ORs) then convert the chemical signals into electrical signals that are"

Please notice this sentence is twice right now: We found that HcunCXEs displayed a either antennae- or leg/wing-biased expression. We found that HcunCXEs displayed either antennae- or leg/wing-biased expression.

M&M
The sentence: "Genes related to the ODEs CXEs of H. cunea and other reported insects of Seasamia..." must be changed to "Genes related to the ODEs CXEs of H. cunea and other reported insects (Seasamia.....)"

Please indicate after "Mafft" that the most suitable evolutionary model was calculated with "X" program and then the phylogenetic tree constructed

Discussion
I do not understand this sentence: “HcunCXE2, 3, 4, 5, 6, 8 and 10 were homologous to this (e.g. DmelCG10175) in D. melanogaster”

Change “In the current study, nine out of 10 HcunCXEs were found to express in the larvae (Fig. 4)” for “In the current study, nine out of 10 HcunCXEs were expressed in the larvae (Fig. 4)”


Figure legends
Genrally secreted enzymes, substrates include hormone and pheromones; B: Generally intracellular enzymes, dietary metabolism/ detoxification functions; C: JHE; D: Genrally secreted enzymes, substrates includes hormone and pheromones; E: Nerouligins; F: ACHE.------------------------------correct “Generally”

Experimental design

Ok

Validity of the findings

Ok

Additional comments

Ok

·

Basic reporting

The authors sufficiently addressed previous comments on the language. The readability of the paper had improved greatly. I have suggested a small number of a single word or sentence structure changes in the attached pdf which may improve the overall clarity and readability of the paper however, these changes are rather minor.

Figure 1.
For clarity the bootstrap values should be represented as percentage or fraction rather than whole integers. For example see PNAS November 12, 1996 93 (23) 13429; https://doi.org/10.1073/pnas.93.23.13429
Is there significance associated with the thickness of the lines for the phylogenetic tree? If yes it should be addressed in the figure description, if not all lines should have the same thickness.
The bootstrap reference should be removed from the figure description or modified.

Experimental design

The manuscript meets all the criteria set out by PeerJ Review. The authors have addressed all issues listed in the previous review.

Validity of the findings

The study is valuable and provides interesting results on the antennal CXEs in H. cunea.
The study provides interesting insights for both the species under investigation as well as the basis for future comparative studies for degradation of Type 1 and Type 2 pheromones.

The conclusions based on the gene expression analysis can be further improved for clarity.
The authors need to combine their results from phylogenetic analysis and expression studies to clearly delineate the candidates for pheromone/volatile degradation and candidates for basic metabolic functions. Please see General Comments section for details.

Additional comments

Overall, this is a very nice paper and a valuable foundation for future work on CXEs. The highlighted portions of the manuscript pdf have associated comments.

I would recommend the following changes to improve the content of the discussion section of the manuscript.


1. Phylogenetic analysis
For A. polyphemus and M. brassica antennal CXE a function as pheromone degradation enzyme has been previously established. If you included the sequences corresponding to these CXEs in your phylogenetic analysis it would be clear that CXE1 and CXE9 are obvious candidates for pheromone/plant volatile degradation. This combined with your gene expression analysis can provide further insights into roles for each of these proteins based on their sex-specific expression patterns.

2. Gene expression analysis
Line 256: You state that "although the sex-biased expression is not
statistically significant, there is a clear numerical difference between expression level in the sexes." However, Figure 2 shows the statistical difference between mean values for CXE1, 3 and 10 and not CXE 9. It is worth to note that the most abundant CXE genes in the antenna are NOT the extracellular CXE that likely participate in the volatile odorant degradation. The most abundant CXE's are likely involved in primary metabolic activities and it would thus make sense their expression is much higher than for the other specialized CXEs in the antenna. Furthermore, some of the CXEs in the metabolic class such as CXE are also highly expressed in other tissues. This is related to your comments in the line 322 where you state that the most highly expressed CXE coding sequences are likely involved in pheromone degradation which is misleading.
Line 273: 1). "We speculated that these H. cunea CXE genes mainly degrade sex pheromone components and host plant volatiles." This is inconsistent with your phylogenetic findings. See note in edited pdf. Only 3 out of the 10 CXE are likely candidates for pheromone degradation. The statement should be revised to reflect that only some of the CXE may putatively participate in pheromone degradation.
Line 278: It is not clear what the authors mean by this statement. It has been previously shown that a single CXE can degrade a number of different pheromones/volatiles and the number of CXEs vary among different insect species.
Line 316-317: It would be beneficial to specifically state here which of the CXEs are antenna specific.
Line 321-325: Need clarification or rewriting.

---

## Round 0.3 · Minor Revisions

Please see my comments on the attached annotated PDF. I look forward to receiving the updated manuscript.

---

## Round 0.4 · accepted · Accept

Thank you for addressing all the minor revisions.